# OpenReview forum: "Normalized Rewards for Preference Learning"
_ICLR.cc/2026/Conference — Submitted to ICLR 2026_

### Official Review · Reviewer_kXKD · 2025-10-24

**Soundness:** 2
**Presentation:** 1
**Contribution:** 2
**Rating:** 2
**Confidence:** 3

**Summary:**

This paper studies over-optimization in Direct Alignment Algorithms (DAAs) like DPO and SimPO.
The authors propose that the over-optimization in implicit reward stems from the neglected partition function's role in normalizing the generation distribution.
The authors introduce a length-normalized probability conservation penalty to mitigate over-optimization.
Empirical results show that the proposed method can reduce over-optimization on some tasks and models.

**Strengths:**

1. The partition function perspective provides an interesting viewpoint on the over-optimization issue in DAAs.
2. The proposed penalty term is a direct and reasonble approach to regularize the sum of probabilities.

**Weaknesses:**

1. **Clarity of Motivation**: The authors argue that over-optimization arises from neglecting the partition function $Z(x)$. However, there exist three different definitions of this partition function in the literature: the DPO & SimPO's partition function to normalize $\pi_\theta \propto \pi_{ref}\exp(r/\beta)$, and the partition function of LLM's softmax output. They are not conceptually equivalent, and it is unclear when the authors argue that the partition function should be $1$ (line 185) and fixed during training.
2. **Lack of Novelty**: The preservation of probability as a regularization term has been explored in prior works, such as adding an SFT loss (which the paper should compare as a baseline).
3. **Unstable Empirical Results**: The empirical results show that the proposed method can reduce over-optimization on some tasks and models, but not consistently across different settings.

**Questions:**

Could you clarify the definition of the partition function you are using, and how it relates to the different definitions in the literature?

---

> ### Author Response · Authors · 2025-11-21
>
> > Clarity of Motivation: The authors argue that over-optimization arises from neglecting the partition function . However, there exist three different definitions of this partition function in the literature: the DPO & SimPO's partition function to normalize , and the partition function of LLM's softmax output. They are not conceptually equivalent, and it is unclear when the authors argue that the partition function should be  (line 185) and fixed during training.
>
> We are happy to clarify the motivation for our regularizer. We take the definition of the partition function Z as the normalization of the policy written in Equation 4. With DPO we are effectively changing the likelihood of our answers to be aligned with preferences. In some cases, DPO updates the model to decrease or increase the likelihood of both responses. Any time the likelihoods of both the chosen and rejected move in the same direction, the partition function is impacted through an increase or decrease in value. Changing the partition function automatically changes the probabilities of all responses, regardless of whether they should have changed given the samples in the current batch. By changing the partition function from a single sample or a small subset of samples, we are changing the probabilities of samples even in cases where this is undesireable. However, if the probability of one response increases while the other decreases for a given preference pair, the partition function is only minimally impacted. This is why we care about not impacting the partition function to avoid unintended consequences (large changes to likelihoods despite no evidence to support these changes) on outliers.
>
> > Lack of Novelty: The preservation of probability as a regularization term has been explored in prior works, such as adding an SFT loss (which the paper should compare as a baseline).
>
> While adding the SFT loss has been considered to address the decreasing likelihoods, we aim to address not only decreasing likelihoods, but more generally, any large change in total likelihood assigned to preferences in the dataset. Adding an SFT loss increases the likelihood of the preferred response, but this is not necessarily desirable, especially if the preferred response is not ideal, and can result in low response diversity. Our regularization term pushes the models towards reallocating likelihood from the rejected response to the preferred response while avoiding increasing the preferred response’s likelihood beyond that. This allows for the regularization term to also help with likelihoods that grow too large, and the regularization applies not only to DPO but methods such as SimPO where over-optimization does not necessarily result in decreasing likelihoods. We agree that adding the SFT loss should be a baseline to compare to and are in the process of running this baseline.
>
>
>
> > Unstable Empirical Results: The empirical results show that the proposed method can reduce over-optimization on some tasks and models, but not consistently across different settings.
>
> We would like to clarify the expected behavior when over-optimization occurs. When the model is over-optimized to a preference dataset, we expect the model to perform well on the dataset distribution, but the model will fail to generalize beyond the distribution and lose some of its initial capabilities. While there are differences in how the method affects performance across different settings, our results demonstrate that by addressing over-optimization, the overall performance across AlpacaEval, common sense and reasoning, and reward modeling generally improves.
>
> > Could you clarify the definition of the partition function you are using, and how it relates to the different definitions in the literature?
>
> We take the definition of the partition function Z as the normalization of the policy written in Equation 4. We will add some clarification in the paper on how to interpret this term below Eq (6) as we agree it is important.

---

### Official Review · Reviewer_7NZV · 2025-10-25

**Soundness:** 2
**Presentation:** 2
**Contribution:** 2
**Rating:** 4
**Confidence:** 3

**Summary:**

This paper presents a well-motivated and technically sound approach to mitigating over-optimization in Direct Alignment Algorithms such as DPO and SimPO by introducing a normalization-based regularization term that stabilizes the implicit reward’s partition function. The proposed methods, N-DPO and N-SimPO, effectively preserve response probability mass, reducing likelihood displacement—particularly from outlier tokens—while improving trade-offs between alignment quality and general reasoning ability. The authors support their claims with solid theoretical reasoning, detailed token-level analyses, and comprehensive experiments on multiple LLMs (Llama-3.1-8B, Mistral-7B, OLMo-7B), showing up to 20% gains on AlpacaEval2 and better benchmark retention. Overall, the work offers a simple yet impactful modification to existing preference optimization methods, enhancing both stability and generalization in post-training alignment.

**Strengths:**

* The paper introduces a clear, theoretically grounded explanation—reward misnormalization—for over-optimization in DAAs, and proposes a simple yet effective normalization regularizer applicable to both DPO and SimPO.

* Extensive experiments across multiple LLMs (Mistral, Llama-3.1, OLMo) show consistent improvements in alignment quality and benchmark performance, supported by detailed token-level analyses.

**Weaknesses:**

* Limited exploration of generality: While results are strong, the experiments are restricted to instruction-following models; broader testing on diverse domains (e.g., reasoning, coding, multimodal tasks) would strengthen claims.

* Hyperparameter sensitivity: The effectiveness of the normalization term depends on tuning λ and β, but the paper offers limited discussion on robustness or practical guidance for real-world adoption.

**Questions:**

1. How sensitive are the proposed N-DPO and N-SimPO methods to the choice of the regularization coefficient λ across different model sizes and datasets?

2. Could the authors clarify whether enforcing a fixed partition function might inadvertently limit expressivity or adaptation in certain preference distributions?

3. Have the authors explored whether the normalization regularizer can be combined with other anti–over-optimization techniques (e.g., token-level confidence weighting or KL-based constraints) for additive benefits?

---

> ### Author Response · Authors · 2025-11-21
>
> > Limited exploration of generality: While results are strong, the experiments are restricted to instruction-following models; broader testing on diverse domains (e.g., reasoning, coding, multimodal tasks) would strengthen claims.
>
> While we do not explore domains such as coding in our paper, as instruction following is a component of reasoning and coding application, we believe our results can generalize to these different domains. Furthermore, we demonstrate that even when training on instruction-following data, applying the regularization term has the benefit not only of maintaining common sense and reasoning performance but also of improving it as seen in Table 2 on page 8. Based on the benefits of the regularization term for common sense and reasoning even without explicit training, we expect the regularization term to similarly help when applied to coding or reasoning tasks.
>
> > Hyperparameter sensitivity: The effectiveness of the normalization term depends on tuning λ and β, but the paper offers limited discussion on robustness or practical guidance for real-world adoption.
>
> Thank you for this feedback, and we will include discussion on the hyperparameters. Adding regularization can affect the optimal beta, but in many cases, the optimal beta remains the same before and after regularization. We found that lambda typically worked best around 0.02 - 0.1. For lambda in the range of 0.25 - 1, we found that performance would more often fall below DPO/SimPO performance. For lambda <0.02, we found that the model performance often improved with growing lambda. As a result, we used the range of [0.025, 0.05, 0.075, 0.1] for lambda and provide the full set of hyperparameters used in the appendix.
>
> > How sensitive are the proposed N-DPO and N-SimPO methods to the choice of the regularization coefficient λ across different model sizes and datasets?
>
> While we did not directly evaluate how lambda changes across settings, we would expect the coefficient to be robust to changes in configurations due to the fact that the general structure of the output probabilities would remain relatively similar across different settings. Additionally, most works on DAAs have performed evaluation on a fixed model size and often in the 7-8B range, and we have focused on the setting consistent with this body of work.
>
> > Could the authors clarify whether enforcing a fixed partition function might inadvertently limit expressivity or adaptation in certain preference distributions?
>
> The proposed partition regularization will limit adaptation for those samples outside the DPO training data distribution. However, we see this as a feature not a bug. The regularizer does not constrain how much the likelihoods of the two responses can change. It instead constrains how much they can change the partition function. Large changes to the partition function will have unintended consequences on samples outside the dataset, which is what we aim to mitigate. Therefore, we are not limiting the extent to which the model can adapt to a given preference dataset. Instead we are limiting how much the preference adaptation affects samples outside the preference dataset. Our results on common sense and reasoning demonstrate that despite strong adaptation to the preference dataset, the model maintains or improves its general capabilities. Additionally, the AlpacaEval results demonstrate the regularizer does not harm generative ability on samples in or near the preference data distribution.
>
> > Have the authors explored whether the normalization regularizer can be combined with other anti–over-optimization techniques (e.g., token-level confidence weighting or KL-based constraints) for additive benefits?
>
> We appreciate this suggestion on potential ways to get additional benefits. We have not explored combining with other techniques as we aimed to primarily validate the effectiveness of the regularization term and explain the mechanisms by which it reduced likelihood displacement. However, we expect that combinations of techniques could provide additional benefits as the regularization term does not provide a fixed change but can be tuned to further improve performance. Additional benefits may be more likely for techniques that do not modify the overall reward shape such as selecting relevant tokens to update.

---

### Official Review · Reviewer_RVXr · 2025-11-01

**Soundness:** 3
**Presentation:** 3
**Contribution:** 3
**Rating:** 4
**Confidence:** 4

**Summary:**

This paper supports a hypothesis that the over-optimization phenomenon in Direct Alignment Algorithms are mostly due to likelihood changes from outlier tokens, which explains why the likelihood of both chosen and rejected responses may decrease simultaneously. To mitigate such effect, they propose a regularization term aiming to maintain the total probability of chosen and rejected responses from shifting. The then show that the proposed approach can mitigate the over-optimization issue (the likelihood of chosen responses increase) and promising improvement in general benchmarks.

**Strengths:**

1. The proposed approach is adding a single regularization term, which could be easily adopted, and it works for both with and without reference algorithms.
2. The paper gives a nice and insightful analysis on the over-optimization phenomenon.

**Weaknesses:**

1. A more justified theoretical analysis on the regularization displacement is preferred. Is there an explanation on why the displacement mainly happens on the outlier tokens?
2. I don't see a clear connection between the partition function and the regularization term. Figure 1 is confusing as there is no explanation on why per-token reward changes in such a way after adding the regularization. I also don't see why keeping the sum of probability of chosen / rejected responses close to the reference policy can lead to a better estimate on the partition function. Can the authors give more explanation / experimental analysis on this?

**Questions:**

See questions above.

---

> ### Author Response · Authors · 2025-11-21
>
> > A more justified theoretical analysis on the regularization displacement is preferred. Is there an explanation on why the displacement mainly happens on the outlier tokens?
>
> Previous work [1] has shown that the gradients are dominated by low likelihood tokens. This could lead to accelerated changes in low likelihood tokens in comparison to other tokens, resulting in the outlier behavior. Additionally, the outliers are more impacted, because there is a lack of evidence and support in the dataset to push their likelihoods in the correct direction. Therefore, when the partition function undergoes large changes over the course of DPO training, the likelihoods of those samples are disproportionately impacted by the partition function changes.  However, due to challenges in describing likelihood dynamics, as the training dynamics of transformers is complex, providing a more precise justification is difficult. Our empirical results support that outlier tokens regularly appear in responses and significantly contribute to likelihood displacement.
>
> [1] Yoon, Hee Suk, et al. "ConfPO: Exploiting Policy Model Confidence for Critical Token Selection in Preference Optimization." Forty-second International Conference on Machine Learning.
>
> > I don't see a clear connection between the partition function and the regularization term. Figure 1 is confusing as there is no explanation on why per-token reward changes in such a way after adding the regularization. I also don't see why keeping the sum of probability of chosen / rejected responses close to the reference policy can lead to a better estimate on the partition function. Can the authors give more explanation / experimental analysis on this?
>
> The introduction of our regularization term prevents the partition from drifting too far from its original value by preventing the likelihood of both responses in a preference pair from increasing or decreasing during the same model update. Instead the regularization encourages the transfer of likelihood (or as reviewer 8RpH terms it, “...conserves the total probability mass...”) between the two responses. This leaves the likelihood function minimally changed, which is the key impact we seek as it mitigates indirect impacts on the probabilities of unseen or underrepresented samples in a given update. We have added a visualization of how our regularization affects the response likelihoods in Figure 2 where we can see adding the regularization term reduces the drop in chosen response likelihoods by over 80%.

---

### Official Review · Reviewer_8RpH · 2025-11-03

**Soundness:** 2
**Presentation:** 3
**Contribution:** 2
**Rating:** 6
**Confidence:** 4

**Summary:**

This paper proposes "Normalized Rewards" to reduce likelihood displacement in Direct Alignment Algorithms (DAAs). The authors frame displacement as a consequence of DAAs implicitly assuming a fixed partition function, which in reality drifts during training. They introduce a regularization term that explicitly conserves the total probability mass of the chosen and rejected responses relative to the reference model.

Question: What if only the winning response probability were regularized with respect to the rejected response. Allow the losing response to drift?

**Strengths:**

### Insightful Partition-function Framing
Connecting likelihood displacement to a ``partition function'' is an elegant framing of the problem. It provides a clean theoretical justification for why generative likelihoods drift than standard explanations based (say based on gradients).


### Valuable Token-level Analysis
Section 4.1 provides a strong contribution by analyzing how displacement happens at the token level. The finding that displacement is often driven by a small set of "outlier tokens" whose probabilities drop sharply is a valuable observation for the community.

EDIT:
I am raising the score further because I really like the log loss square formulation. In fact it gives a different effect than p log p. The reason is that it overweights the lower values I believe? Can you add some thoughts on difference between this and entropy (plogp)?

**Weaknesses:**

**Missing plot of the main claim**
The abstract explicitly states that DAAs "decrease the likelihood of preferred responses" and implies this method fixes it. While Figure 1 provides a high-level schematic and Figure 2 shows token-level distributions at a single snapshot, the paper lacks a direct "before-and-after" training curve. It would be great to see a plot of Average LogProb(Preferred) vs Training Steps for both DPO and N-DPO to confirm that the proposed regularization actually stabilizes this metric during training.

**Recent methods missing**
The paper benchmarks against methods that achieve ~30% win rates, while recent work from late 2024 and 2025 has higher performance on the same model class. Does the regularizer here port to these settings? Does it still improve them? While I would like to see these results, at the least they should be referenced as they are now a part of literature in this area.

**Simpo Baseline unclear**
The reported SimPO baseline for Llama-3-8B (32.79% LC WR) is significantly lower than the ~44% reported in the original SimPO (see Table 1 of that paper). Furthermore, Table 1 shows N-SimPO underperforms this (31.01%).

---

### References

[1] Chen, H., He, G., Yuan, L., Cui, G., Su, H., & Zhu, J. (2024). Noise contrastive alignment of language models with explicit rewards. Advances in Neural Information Processing Systems, 37, 117784-117812.

[2] Wu, Y., Sun, Z., Hughes, R., Ji, K., Yang, Y., & Gu, Q. (2025). Self-play preference optimization for language model alignment., International Conference on Representation Learning (Vol. 2025, pp. 91558–91582).

[3] Gupta, T., et al. (2025). AMPO: Active Multi Preference Optimization for Self-play Preference Selection. ICML 2025.

[4] Gupta, T., et al. (2025). REFA: Reference Free Alignment with Fine-Grained Length Control. COLM 2025.

[5] Tang, X., et al. (2025). Game-Theoretic Regularized Self-Play Alignment of Large Language Models. arXiv preprint arXiv:2503.00030.

**Questions:**

1. Can you concretely add the loss function used to the draft somewhere? Not just the regularizer?
2. Your regularizer targets the sum: $\log([\pi_\theta(y_w) + \pi_\theta(y_l)] / [\pi_{ref}(y_w) + \pi_{ref}(y_l)])^2$. I suspect many of the gains might come simply from preventing $y_w$ from drifting. Have you tried regularizing just the winning response: $\log([\pi_\theta(y_w)] / [\pi_{ref}(y_w)])^2$? If this simpler version works equally well, the "partition function" motivation (which relies on the sum representing the whole space) might be less relevant than simple SFT-style regularization.
On the other hand, if this simpler version does not work, it would be strong motivation of why the sum, or partition function is necessary.
3. Apart from 2, a natural variant you may consider is also $\mathcal{R} = \lambda [(\log \pi_\theta(y_w) - \log \pi_{ref}(y_w))^2 + (\log \pi_\theta(y_l) - \log \pi_{ref}(y_l))^2]$. But i suspect you should outperform this.
4. N-SimPO hurt performance on your strongest model, so a possible conclusion is that you used too high a lambda. Can you reduce it, and show 2-3 more values for a lambda ablation?

---

Can you provide training curves showing Average LogProb(Preferred) vs Steps for DPO and N-DPO? Visual proof that your method stabilizes this metric over time or at least improves this metric over the baseline DPO, along with some of the above experiments suggested, and fixed references would be good.

---

> ### Author Response · Authors · 2025-11-21
>
> > Can you add some thoughts on difference between this and entropy (plogp)?
>
> We would be happy to share thoughts on differences between our regularization function and entropy. Assuming that p is the response likelihood for the reference model and q is the response likelihood for the current model, if we consider the gradient of $p \log q$, we would have $p \nabla_\theta \log q$ while for our regularization term, we would have $2(\log p - \log q) \nabla_\theta \log q$. Since $\log p$ does not change with $\theta$, we have that if we use entropy, the size of the gradient of the regularization term would be mostly independent of how far the current model is from the reference model. If entropy could be estimated based on a more representative set of samples, it may be an effective regularizer, but since preference learning is often done with pairwise comparisons, entropy can be less effective.
>
> > Missing plot of the main claim
>
> We have added a plot in Figure 2 of the average log probs for both preferred and rejected responses over the course of training for DPO, N-DPO, SimPO, and N-SimPO. We show that with regularization, the likelihood drops less while maintaining or increasing the difference in likelihood between chosen and rejected responses. For DPO, the addition of the regularization term reduces the drop in the chosen response likelihood by over 80%.
>
> > Recent methods missing
>
> We appreciate the references to the other methods and have added discussion around these recent methods such as REFA in our related works. We focused on DPO and SimPO in particular as many recent DAAs build on top of DPO and SimPO. Furthermore, since methods such as alpha-PO aim to mitigate similar issues, we focused on adding the regularization term to methods without such mitigation. As a result, we also expect adding the regularizer to other more recent methods to potentially better address likelihood displacement and over-optimization.
>
> > Simpo Baseline unclear
>
> The results in the original SimPO paper are based on a dataset with responses sampled from the reference model, making them on-policy which has been shown to improve performance. We directly apply training to existing datasets such as UltraFeedback, which is off-policy, and use this dataset across all settings in our paper. Additionally, when selecting our training hyper-parameters, we conducted a hyper-parameter sweep per method and chose the best performing hyper-parameters per method. We believe using a consistent dataset and selecting hyper-parameters per method provides a fair comparison.
>
> > Can you concretely add the loss function used to the draft somewhere? Not just the regularizer?
>
> We provide the full loss function in equations 13 and 14. Please let us know if you would like further elaboration on equations 13 and 14.
>
> > Have you tried regularizing just the winning response?
>
> We have not tried regularizing just the winning response, but we expect the behavior of this to be similar to adding an SFT term as it only regularizes the likelihood of one response. Adding the SFT loss has been considered to address the decreasing likelihoods and should prevent large negative changes to the winning response’s likelihood and prevent changing the partition function too much. However, our worry is it does not prevent the winning response from having a large likelihood increase and changing the partition function. We aim to address not only decreasing likelihoods but more generally, any large change in total likelihood assigned to response pairs in the dataset and the resulting impact on the partition function. Similarly, regularizing only the winning response to be fixed may not necessarily maintain the total response likelihood. Our worry is it does not prevent the losing response from drifting and changing the partition function. We are currently in the process of running a comparison with SFT as a regularizer and can share further insights based on the results.

---

> > ### Author Response · Authors · 2025-11-21
> >
> > > Apart from 2, a natural variant you may consider is also $\lambda [(\log \pi_\theta (y_w) - \log \pi_{ref}(y_w))^2 + (\log \pi_\theta (y_l) - \log \pi_{ref}(y_l))^2]$.
> >
> > Our interpretation of the regularizer you have proposed is that it will limit the extent to which the implicit reward is able to fit the training data. This is especially true in cases where the rejected response is initially more likely as the variant would prevent this example from flipping. We agree this should limit changes to the partition function, however, it also appears to result in underfitting the training data. For this reason, we focused our explorations on regularizers that conserved the total probabiltiy while allowing both likelihoods to change. Our regularization term does not aim to fix the likelihoods of individual responses, but instead reallocate the total probability assigned to chosen and rejected responses in a way that makes the chosen responses more likely.
> >
> > > N-SimPO hurt performance on your strongest model, so a possible conclusion is that you used too high a lambda. Can you reduce it, and show 2-3 more values for a lambda ablation?
> >
> > While N-SimPO does result in a slight decrease in AlpacaEval scores, it increases common sense and reasoning benchmark scores by over 2.5% on average compared to a lambda of 0. Reducing lambda would result in model performance in between that of the current N-SimPO model and the SimPO model. As a result, reducing lambda may slightly increase AlpacaEval scores, but it could also reduce common sense and reasoning performance. The current value of lambda is also chosen based on AlpacaEval1.0 performance out of a hyperparameter range of [0.025, 0.05, 0.075, 0.1].

---

### Author Response · Authors · 2025-11-21

We thank all reviewers for their time considering our submission, and are happy to see that reviewers found N-DPO/N-SimPO as having an **insightful partition-function framing**, which provides an **interesting viewpoint** on likelihood optimization, is **theoretically grounded**, and **could be easily adopted**. Reviewers have also stated that our experiments already provide **insightful analysis** on over-optimization, and **extensive experiments** across LLMs.

We would like to summarize some of the main points that we have addressed in the response:

- **Confusion on explanation of N-DPO/N-SimPO** – We updated **Figure 1** and added clarifications after Eq. 6 to provide a clearer intuition for the method and directly show the reduced likelihood displacement in **Figure 2**. We motivate the regularizer as a means to keep changes to the partition function minimal to avoid large changes in the likelihood of responses that are irrelevant to the preference data while also allowing the implicit preference model to fully capture the training data.

- **Hyperparameter sensitivity** - We provide clarification on hyperparameter sensitivity, in particular, the optimal beta remains the same in many cases with and without regularization.

- **Comparisons with other methods** – We are in the process of providing comparisons with other methods for reducing likelihood displacement such as adding an SFT term. Our regularization can also be used and tuned to work on top of other methods as well.

---

### Meta-Review · Area_Chair_XuV9 · 2026-01-08

**Summary:**

The paper investigates "likelihood displacement" in Direct Alignment Algorithms (DAAs), proposing that reward misnormalization stems from a failure to account for the partition function during training. To address this, the authors introduce a regularization term intended to conserve the total probability mass of chosen and rejected responses.

**Reviewer Concerns:**

While the reviewers initially appreciated the "partition-function framing" and the "token-level analysis," the consensus for rejection is driven by significant concerns regarding conceptual clarity, empirical consistency, and marginal novelty. Specifically, the committee found the theoretical connection between the transformer's softmax output and the implicit DPO reward's partition function to be insufficiently substantiated. Furthermore, the empirical results were viewed as inconsistent across different model architectures (e.g., Llama vs. OLMo), and the proposed regularization was noted to share substantial conceptual overlap with existing techniques, such as SFT-objective augmentation, without demonstrating a transformative advantage. The authors' rebuttal, while providing additional training curves, failed to resolve the fundamental ambiguity regarding why this specific formulation is theoretically superior to simpler probability-based constraints.

**Reviewer Scores:**

A clear consensus for a negative recommendation has emerged among the reviewers, as significant concerns remain unaddressed despite the rebuttal.

---

### Decision · Program_Chairs · 2026-01-26

Reject